Subject Area:
neuroscience/molecular biology/genetics/cellular biology/biochemistry

Keywords:
Wallerian degeneration, injury-induced axon degeneration, axon death, neurodegeneration

Author for correspondence:
Lukas J. Neukomm
e-mail: lukas.neukomm@unil.ch

# Axon death signalling in Wallerian degeneration among species and in disease

Arnau Llobet Rosell and Lukas J. Neukomm

Department of Fundamental Neurosciences, University of Lausanne, Rue du Bugnon 9, 1005 Lausanne, VD, Switzerland

ALR, 0000-0001-7728-2999; LJN, 0000-0002-5007-3959

Axon loss is a shared feature of nervous systems being challenged in neurological disease, by chemotherapy or mechanical force. Axons take up the vast majority of the neuronal volume, thus numerous axonal intrinsic and glial extrinsic support mechanisms have evolved to promote lifelong axonal survival. Impaired support leads to axon degeneration, yet underlying intrinsic signalling cascades actively promoting the disassembly of axons remain poorly understood in any context, making the development to attenuate axon degeneration challenging. Wallerian degeneration serves as a simple model to study how axons undergo injury-induced axon degeneration (axon death). Severed axons actively execute their own destruction through an evolutionarily conserved axon death signalling cascade. This pathway is also activated in the absence of injury in diseased and challenged nervous systems. Gaining insights into mechanisms underlying axon death signalling could therefore help to define targets to block axon loss. Herein, we summarize features of axon death at the molecular and subcellular level. Recently identified and characterized mediators of axon death signalling are comprehensively discussed in detail, and commonalities and differences across species highlighted. We conclude with a summary of engaged axon death signalling in humans and animal models of neurological conditions. Thus, gaining mechanistic insights into axon death signalling broadens our understanding beyond a simple injury model. It harbours the potential to define targets for therapeutic intervention in a broad range of human axonopathies.

## 1. Introduction

Neurons use their axons to communicate with remote cells. These axons can be extremely long, ranging from millimetres to centimetres to metres depending on the host, the type of neuron and the target cell [1,2]. In other words, axons can take up more than 99.9% of the neuronal volume. Axons also harbour a remarkably elaborate axonal cytoskeleton (axoskeleton) [3], which allows them to withstand stretch, compression, tension and torsion [4–8]. To ensure continued circuit function, the nervous system established soma-independent, local axonal-intrinsic and glial-extrinsic mechanisms to support lifelong axon survival [9]. If these survival mechanisms are impaired, axons will undergo axon degeneration [10,11]. Axonopathies are increasingly recognized as major contributors in neurological conditions, such as Alzheimer's disease (AD), amyotrophic lateral sclerosis (ALS) and multiple sclerosis (MS) [12], Parkinson's disease (PD) [13], traumatic brain injury (TBI) [14], and chemotherapy-induced peripheral neuropathy (CIPN) [15]. Axon degeneration occurs prior to neuronal loss in a broad range of injured and diseased nervous systems, thus targeting it by therapeutics serves as a promising opportunity to ameliorate neurological disorders [16,17]. Gaining insights into underlying mechanisms executing axon degeneration will help to

royalsocietypublishing.org/journal/rsob   Open Biol. **9**: 190118

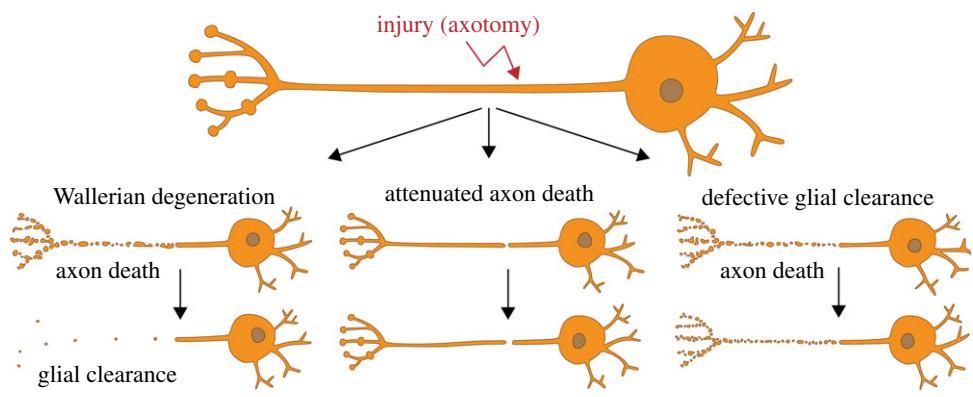

**Figure 1.** Wallerian degeneration consists of two molecularly distinct programmes. Upon axotomy, the axon separated from the soma actively executes its own fragmentation (axon death), which is mediated by an evolutionarily conserved axon death signalling cascade. The severed axon undergoes axon death within 1 day. Surrounding glial cells will then engage and clear the resulting axonal debris within 3–5 days. Attenuated axon death results in severed axons which remain functionally and morphologically preserved for weeks to months, while defective glial clearance culminates in axonal debris which persists for a similar time *in vivo*.

define targets for the development of efficacious drugs for therapeutic intervention. However, how axon degeneration is mechanistically regulated and executed remains currently largely unknown.

To date, distinct morphological modes of axon degeneration have been observed and underlying molecular mechanisms described [10]. Among them are dying back axon degeneration, retraction, axosome shedding, focal axonal degeneration induced by growth factor withdrawal and pruning, to name a few. Axon degeneration can also be triggered through axonal injury (axotomy), which is probably one of the simplest models to study how axons execute their own destruction. Identified by and named after Augustus Waller, Wallerian degeneration (WD) is an umbrella term under which two distinct mechanisms occur [18] (figure 1): first, severed axons—separated from the soma—actively execute their own disassembly (axon death) within 1 day, through an evolutionary conserved axon death signalling cascade; and second, surrounding glial cells engage and clear the resulting debris within 3–5 days. Axon death and glial clearance are separate processes (figure 1): the attenuation of axon death results in severed axons which remain functionally and morphologically preserved for weeks to months, while defective glial clearance culminates in axonal debris which persists for a similar time *in vivo* [19,20].

Axon death signalling is activated not only when the axon is cut, crushed or stretched [21,22], it also seems to be a major contributor in different animal models of neurological conditions, e.g. where axons degenerate in the absence of injury [23]. Therefore, axon death signalling could provide insights into axon degeneration beyond a simple injury model or, as August Waller wrote [18], 'It is impossible not to anticipate important results from the application of this inquiry to the different nerves of the animal system. But it is particularly with reference to nervous diseases that it will be most desirable to extend these researches'.

Over recent years, significant advancements broadened our understanding of axon death signalling in WD. While these discoveries shed light onto distinct mediators of axon death, they also led to more unanswered questions. First, how complete is our understanding of axon death signalling? Second, what differences or commonalities have been observed across species? And third, how do these findings bring us closer to defining new pharmacological targets

that could prevent axon degeneration in the diseased and injured nervous system?

In this review, we aim to provide answers to the above questions. First, we will comprehensively summarize our current understanding of axon death signalling across species. Second, we will highlight differences observed across model systems and discuss a potential core mechanism shared across species. And third, we will conclude with a brief overview of engaged axon death signalling in the diseased or injured nervous system.

## 2. Phases of Wallerian degeneration

Immediately after axotomy, the axon that has been separated from the soma goes through a lag phase where its overall morphology remains unchanged for 6–24 h depending on a number of factors, such as *in vitro* or *in vivo* models, and animal systems used (figure 2*a*). During this phase, the earliest feature observed immediately after injury is a rapid, short-term increase of axonal calcium ($Ca^{2+}$) levels in both the proximal and distal axon stump (figure 2*b*). This is mostly due to extracellular $Ca^{2+}$ influx at the lesion site, and to a lesser extent from axon internal $Ca^{2+}$ stores [24–27]. After the first, rapid short-term $Ca^{2+}$ wave, nicotinamide adenine dinucleotide ($NAD^+$) and adenosine triphosphate (ATP) are rapidly depleted, thus impairing axonal energy homeostasis [28–30]. Mitochondria lose their membrane potential and begin to swell, thereby increasing the generation of reactive oxygen species (ROS). Ultimately, they release their internal $Ca^{2+}$ stores, which culminates in a second, long-term $Ca^{2+}$ wave [28,31–33]. At this point, the gross axonal morphology remains unchanged, despite the already initiated destabilization of microtubules [28,29].

Suddenly, the execution phase starts (figure 2): the axon begins to disassemble. Catastrophic granular fragmentation is observed at the molecular, ultrastructural and morphological level [28]. Microtubules start to disrupt, alongside the dismantling of the axoskeleton. The axon starts 'beading' or swelling, culminating in catastrophic axonal fragmentation [30,34,35].

During the execution phase, surrounding glial cells and specialized phagocytes not only clear the resulting axonal debris by activating multiple signalling pathways [19,36–39], but also actively enhance axonal fragmentation [40]. To date,

royalsocietypublishing.org/journal/rsob Open Biol. 9: 190118

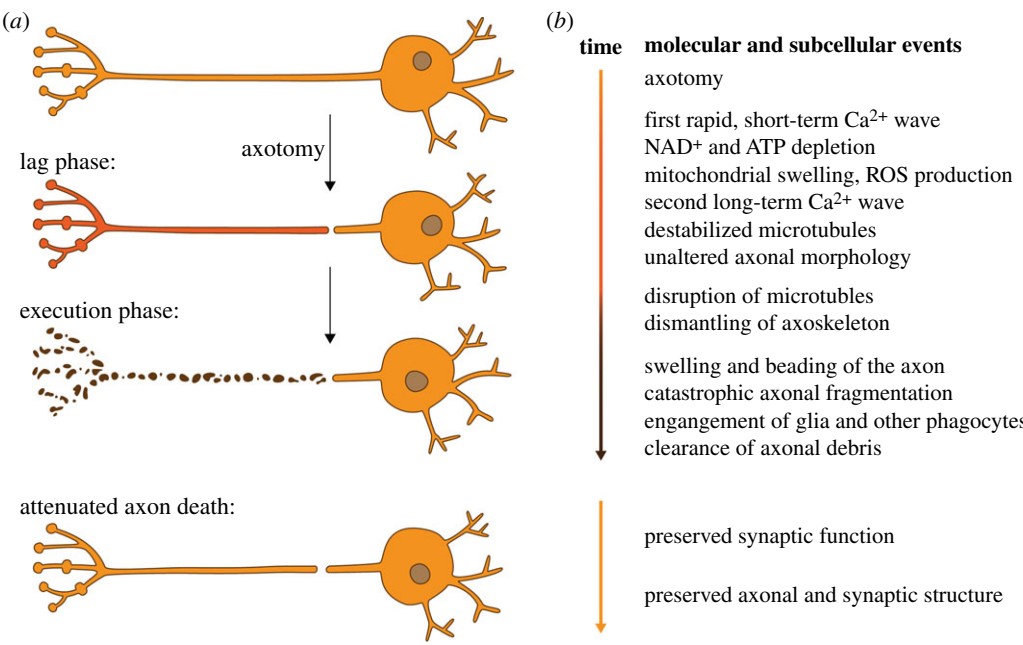

**Figure 2.** Phases and events during Wallerian degeneration. (*a*) Morphological phases of Wallerian degeneration. After axotomy, the axon that is separated from the soma goes through a lag phase (red), where its morphology remains grossly intact. During the execution phase (brown), the axon undergoes catastrophic axon fragmentation (axon death). Surrounding glia engage and clear the resulting axonal debris. The attenuation of axon death signalling results in morphologically and functionally preserved axons and synapses. (*b*) Molecular and subcellular events during Wallerian degeneration. Axotomy, lag and execution phase (orange, red and brown, respectively). The precise onset and duration of each event depends on the model system used. Attenuated axon death signalling preserves the structure and function of axons and their synapses.

several key questions remain unresolved, such as what kind of axonal 'eat me' signals are presented to surrounding glia [41], and where within the execution phase glial cells engage to clear axonal debris.

Once the execution phase is over, e.g. when the injured axon is disassembled and the resulting debris cleared by surrounding glia, WD is also over. While WD occurs in both the central and peripheral nervous system (CNS and PNS, respectively), it is of particular importance in the PNS to pave the way for the proximal axon still attached to the neuronal soma, which ultimately activates its regenerative programme, to regrow and thereby to re-establish circuit function [33].

# 3. The discovery and implications of Wld$^S$

Axon death within WD was long thought to be a simple passive wasting away of severed axons [18]. In 1989, the serendipitous discovery of the 'Wallerian degeneration slow' (Wld$^S$) mouse challenged this idea. Severed Wld$^S$ axons remained preserved for weeks rather than undergoing axon death within a day (figure 2) [42]. Subsequently, the molecular change in Wld$^S$ mice was identified as a tandem triplication of two neighbouring genes: the N-terminal 70 amino acid fragment of the ubiquitination factor E4B (Ube4b) fused to full length nicotinamide mononucleotide adenylyltransferase 1 (Nmnat1) [43]. This fusion results in a translocation of nuclear Nmnat1 to the axon, where it exerts axon death attenuation (this will be discussed further below). The Wld$^S$ neomorph provided the first evidence that axon death could be an active self-destruction programme to eliminate damaged axons; that is, because this programme is profoundly attenuated by the sole over-expression of Wld$^S$. Following the

identification of Wld$^S$, several important discoveries were made, which are summarized below.

First, Wld$^S$ harbours an evolutionary conserved function. While it was first found to delay axon death in mice [42,43], the mouse chimeric protein performs its function equally well in rats [44], fruit flies [19] and zebrafish [33,45]. Wld$^S$ is therefore capable of delaying axon death across multiple species.

Second, Wld$^S$ acts autonomously in neurons and is dosage dependent. The Wld$^S$ mouse harbours a tandem triplication which results in over-expressed Wld$^S$ [43]. The higher the levels, the better its capability to preserve severed axons [43].

Third, Wld$^S$ acts locally within axons. While abundant in both nuclei and axons, Wld$^S$ confers potent attenuation of axon death signalling specifically in axons [46]. Viral transduction of multiple modified versions of Wld$^S$ into severed axons up to 4 h after axotomy (e.g. early within the lag phase) is sufficient to attenuate axon death signalling [47].

Fourth, Wld$^S$ is specific for axon death. Its expression blocks axon death and seems to have largely no effect on developmental pruning [48], or other types of neuronal death such as apoptosis [49].

And fifth, Wld$^S$ is beneficial in many models of neurodegenerative conditions. Since its discovery, the Wld$^S$ mouse has been crossed into a vast number of different neurological models to assess its protective ability in the injured and diseased nervous system [23]. These findings support the idea that axon death is shared in injury and disease, and, more importantly, that axon degeneration is a major driver in different neurological conditions.

Wld$^S$ provided the first evidence that, at the molecular level, axon death can be attenuated by over-expression of a single protein. Its discovery and characterization raised two key questions: first, how is Wld$^S$ able to attenuate an axonal-intrinsic axon death signalling pathway actively executing the degeneration

**4**

royalsocietypublishing.org/journal/rsob    *Open Biol.* **9**: 190118

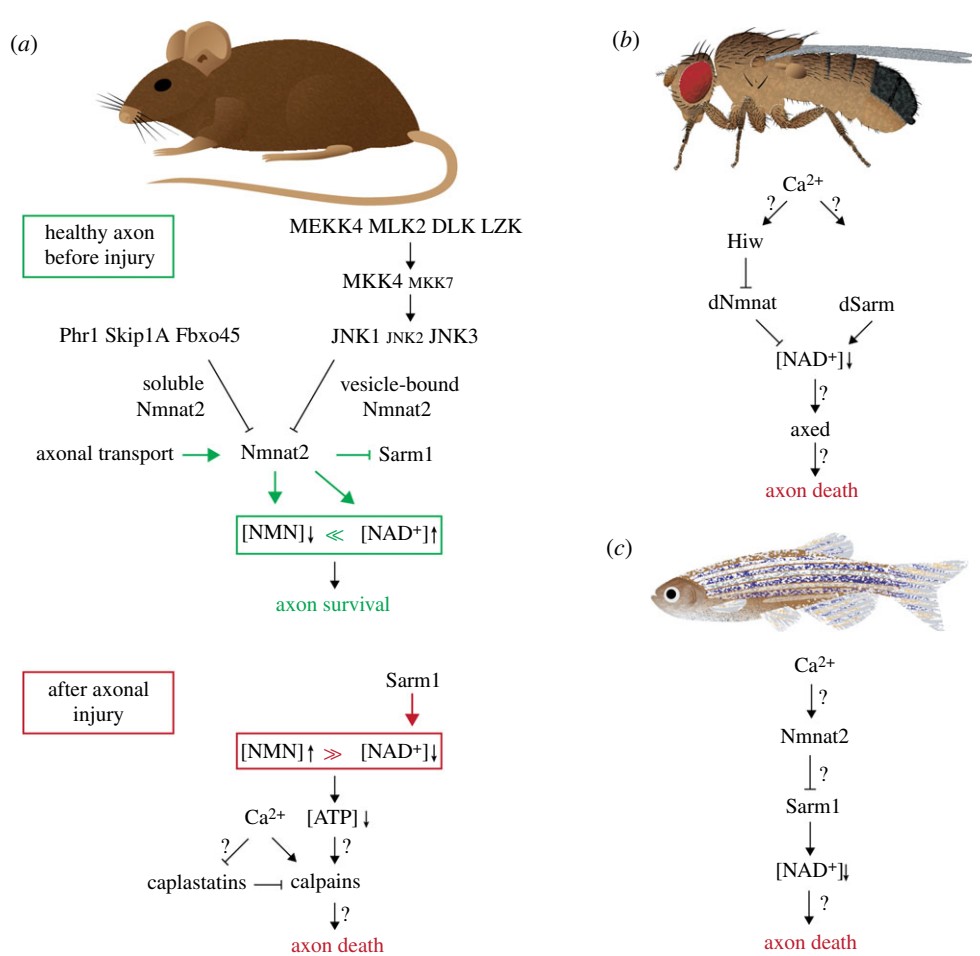

**Figure 3.** Axon death signalling across species. (*a*) In mouse axons, axonal survival is ensured by above-threshold levels of Nmnat2. Nmnat2 is constantly replenished by axonal transport, while the atypical ubiquitin ligase complex and the MAPK pathway rapidly degrade soluble and vesicle-bound Nmnat2, respectively. Nmnat2 keeps NMN levels low, and NAD$^+$ levels constant. Nmnat2 also blocks Sarm1 through an unknown mechanism. Upon axotomy, Nmnat2 drops below threshold levels, which induces axon death. NMN levels rise, and NAD$^+$ levels are rapidly depleted by the NADase activity of Sarm1. (*b*) Axon death signalling in flies. See text for details. (*c*) Axon death signalling in fish. The genetic interactions of Ca$^{2+}$, Nmnat2 and Sarm1 remain to be determined. See text for details.

of severed axons; and second, what are the mediators of axon death signalling? The next section will summarize our current knowledge related to these two questions.

# 4. Axon death mediators

To date, axon death in WD is observed in several species such as mice, rats, flies and fish (figure 3). So far, each animal model has contributed to the discovery of important axon death mediators and also offered distinct insights into axon death signalling. Importantly, the modification of each mediator attenuates axon death comparable to the effects of Wld$^S$, e.g. in the range of days to weeks.

## 4.1. Ca$^{2+}$

As mentioned above, a local initial influx of Ca$^{2+}$ right after axotomy from the site of injury is observed in both invertebrate and vertebrate models *in vivo* and *in vitro* [24,27,50,51]. The first, rapid short-term Ca$^{2+}$ wave precedes the lag phase (figure 2). The disassembly of severed axons is initiated by high extra-axonal Ca$^{2+}$ concentrations: if the extracellular environment is devoid of Ca$^{2+}$ (by adding EGTA, or in a medium lacking Ca$^{2+}$), or when voltage-gated Ca$^{2+}$ channels are inhibited, the fragmentation of severed axons is significantly delayed (figure 3a) [26,50,52–54].

The second, long-term Ca$^{2+}$ wave is present solely in the separated axon. It appears prior to the onset of axonal fragmentation, and is believed to be a key instructive component for the execution phase [51,55]. During the second wave, Ca$^{2+}$ is released from intra-axonal stores (e.g. mainly from mitochondria, and to a lesser extent from the endoplasmic reticulum), and the inhibition thereof can significantly delay the disassembly of the axon [25].

The expression of Wld$^S$ in zebrafish and rodent neuronal cultures largely suppresses the second, but not the first, Ca$^{2+}$ wave, suggesting that the second wave is the one responsible for triggering axon fragmentation [24,51]. In *Drosophila* larvae, Wld$^S$ is also suppressing the first Ca$^{2+}$ wave, reflecting minor differences between the experimental systems [27].

Taken together, the rise in intra-axonal Ca$^{2+}$, both during the immediate first short-term as well as during the slower and delayed second long-term Ca$^{2+}$ wave, is an important instructive signal to trigger axon death signalling. However, how transient high Ca$^{2+}$ levels are linked to the initiation of axon death signalling remains completely unknown.

## 4.2. Nmnat2/dNmnat

Wld$^S$ is an over-expressed version of Nmnat1 translocated from the nucleus to the axon [43,56]. Nmnat proteins are essential for ATP-dependent NAD$^+$ synthesis from either

royalsocietypublishing.org/journal/rsob  Open Biol. 9: 190118

nicotinamide mononucleotide (NMN) or nicotinic acid mononucleotide (NaMN) [57]. Mammals and zebrafish harbour three different Nmnat genes (Nmnat1–3) with different subcellular locations and kinetic properties [58]. In contrast, *Drosophila* relies on a single *nmnat* gene (*dnmnat*) [38], which is alternatively spliced to generate variants localized to the nucleus and the cytoplasm [59]. The discovery of Wld$^S$, and subsequently the extensive research on Nmnat proteins and their biosynthetic NAD$^+$ activity, provided the following four major insights: levels and localization of Nmnat proteins as well as levels of NAD$^+$ and its precursor NMN are all crucial for the execution of axon death signalling.

### 4.2.1. Nmnat2/dNmnat levels

Among the three mammalian Nmnat isoforms, Nmnat2 is the gatekeeper of axon death: it is a limiting, labile axon survival factor [60]. Synthesized in the soma and transported into the axon, Nmnat2 has to be constantly replenished in axons due to its rapid turnover (figure 3*a*): Nmnat2 is the most labile member of its family with a half-life of less than 4 h, despite being the most highly expressed isoform in the brain [61]. Upon axotomy, Nmnat2 fails to be transported to the axon; thus, in severed axons, Nmnat2 levels drop rapidly, and axon death is initiated [60]. Vice versa, depletion of Nmnat2 in whole neurons is sufficient to induce axon degeneration in the absence of injury: axons undergo Wallerian-like degeneration—because they can be attenuated by Wld$^S$—while the soma remains unaffected [60,62]. Similarly, *Nmnat2*$^{-/-}$ mice harbour perinatal lethality due to limited axon extensions, and their isolated neuronal cultures from both PNS and CNS contain neurite outgrowth consistently stalling at 1–2 mm [63]. Likewise, in *Drosophila*, RNAi-mediated knockdown, or a lack of dNmnat in mutants, leads to spontaneous axon degeneration prior to cell body degeneration [64,65]. In zebrafish, over-expression of either Wld$^S$ or Nmnat2 results in a potent attenuation of axon death *in vivo* [33,45]. Whether knock-down of Nmnat2 also results in Wallerian-like degeneration remains to be determined. Taken together, levels of mammalian, fish and fly Nmnat2/dNmnat are crucial for axonal survival (figure 3). It is important to note that any engineered version of stabilized Nmnat protein is capable of attenuating axon death in a variety of different models, e.g. Wld$^S$ as non-nuclear Nmnat1, Nmnat1 that fails to localize to nuclei (e.g. cytosolic and axonal Nmnat1, cyt-Nmnat1 and ax-Nmnat1, respectively), Nmnat2 able to persist longer in axons, *Drosophila* dNmnat over-expressed in cultured mouse dorsal root ganglia (DRG) or mouse Nmnat1 in *Drosophila* olfactory receptor neurons (ORNs) [45,61,62,66–68]. Therefore, above-threshold levels of Nmnat2 or modified versions thereof ensure the survival of the axon, while sub-threshold levels trigger axon death.

### 4.2.2. Nmnat2/dNmnat localization

Nmnat2 is predominantly found in cytoplasm and axoplasm, and is associated with membranes of Golgi-derived transport vesicles undergoing fast axonal transport [63]. Both the removal of the vesicle-association domain in Nmnat2 [62] or the modification of its residues required for palmitoylation [69] lead to increased Nmnat2 half-lives. Moreover, in mammals or flies, the over-expression of Nmnat3, which is predominantly found in mitochondria in mammals, potently attenuates axon death

[67,70]. Therefore, the subcellular localization dictates turnover rates of Nmnat proteins. By targeting them to different axonal compartments relative to endogenously localized Nmnat2, their turnover is reduced, which results in increased Nmnat protein half-lives, and ultimately in attenuated axon death.

### 4.2.3. Nmnat substrate: nicotinamide mononucleotide (NMN)

The attenuation of axon death signalling requires NAD$^+$ biosynthetic activity of Nmnat proteins [33,61,67,71,72]. Metabolic aspects of NAD$^+$ in neurodegeneration and axon death signalling have recently been comprehensively reviewed in [57]. Importantly, genetic and pharmacological modifications of NAD$^+$ metabolism result in different levels of attenuation [73]. Here, we will focus on manipulations that result in a robust long-term (days to weeks) or intermediate (2–3 days) attenuation of axon death signalling, respectively. However, we do note that there are manipulations which result in a short-term attenuation, e.g. a delay of axon death signalling in the range of a day. These observations remain subject to further analyses and will not be covered here.

Nmnat proteins use either NMN or NaMN to generate NAD$^+$ in an ATP-dependent manner. After injury, Nmnat protein levels drop, resulting in increased Nmnat substrates, e.g. NaMN and/or NMN, and decreased Nmnat products, e.g. NAD$^+$. But which one is important for axon death signalling?

While levels of NaMN seem to be negligible [73], levels of NMN are crucial: they temporally rise within 6 h in severed axons and seem to be an instructive signal [55,74]. Preventing this NMN rise results in long-term attenuation of axon death: either by the expression of Wld$^S$ or modified Nmnat proteins that consume NMN to generate NAD$^+$ or—interestingly—by the expression of a bacterial specific enzyme, NMN deamidase, which consumes NMN but does not generate NAD$^+$, both *in vitro* [55,73,74] and *in vivo* in zebrafish and mice [75].

There is an alternative modification to keep NMN levels low: nicotinamide (Nam) is consumed by nicotinamide phosphoribosyltransferase (NAMPT) to generate NMN. The pharmacological inhibition of NAMPT by FK866, either right before or immediately after injury, prevents the rise of NMN, which results in short-term to intermediate attenuation of axon death signalling [73,74].

In summary, distinct pharmacological or genetic manipulations that lead to low levels of NMN also prevent axon death. Thus, the temporal rise of NMN after injury seems to be an instructive signal for axons to execute their own destruction.

### 4.2.4. Nmnat product: nicotinamide adenine dinucleotide (NAD$^+$)

The involvement of NAD$^+$ as a product of the biosynthetic activity of Nmnat proteins in axon death was observed right after the discovery of Wld$^S$: before or immediately after axonal injury, the exogenous supply of high levels of NAD$^+$ leads to long-range attenuation of axon death signalling [71,72]. Thus, axon death is triggered by NAD$^+$ depletion in severed axons. It is important to note that Wld$^S$ or engineered Nmnat proteins do not generate higher levels of NAD$^+$, they solely prevent the depletion thereof by a—yet unknown—mechanism that inhibits an NAD$^+$ consuming enzyme, which will be discussed further below [73,76].

Wld$^S$ got its foot in the door of axon death signalling and paved the way for several key discoveries related to Nmnat proteins. Levels and localization of the axonal survival

factor Nmnat2/dNmnat are crucial for the survival or the degeneration of the axon. Its biosynthetic activity keeps levels of NMN as substrate low, and prevents the depletion of NAD$^+$ as product (figure 3a).

Given the essential role of Nmnat proteins in axon death, an important question is how their turnover and half-life is regulated. What are the mechanisms regulating protein levels and therefore biosynthetic activity? Below, we will discuss the mechanisms that are crucial for Nmnat2 protein levels.

## 4.3. Atypical ubiquitin ligase complex

Following the cloning of Wld$^S$ [43], the ubiquitin proteasome system (UPS) was also found to be involved in the early stages of axon death [29]. The subsequent discovery of Nmnat2 as a labile axon survival factor, which is subjected to rapid turnover in axons [60], led to an attractive hypothesis: could the UPS be responsible for the rapid turnover of Nmnat2 in axons? The first evidence for the UPS system to be involved in axon death was found in *Drosophila*. The E3 ubiquitin ligase Highwire (Hiw) regulates the turnover of dNmnat [20,77]. Likewise, the mammalian homologue PAM/Highwire/RPM-1 (Phr1) also fine-tunes levels of Nmnat2 [78]. Phr1 belongs to an evolutionarily conserved, atypical Skp/Cullin/F-box (SCF)-type E3 ubiquitin ligase complex consisting of S-phase kinase-associated protein 1A (Skip1a), Phr1 and F-box protein 45 (Fbxo45) [78,79]. This atypical SCF complex regulates—through polyubiquitination and the proteasome—levels of Nmnat2, by specifically targeting axoplasmic Nmnat2 for destruction [69,80]. The removal of either component slows down the turnover of Nmnat2, which results in attenuated axon death signalling (figure 3a).

## 4.4. MAPK signalling

The mitogen-activated protein kinase (MAPK) signalling pathway is also involved in axon death signalling. It is activated within 5 min after axonal injury and culminates in the phosphorylation of c-Jun N-terminal kinases (JNKs): loss-of-function analyses revealed that a partially redundant MAPK cascade is required to execute the degeneration of axons after injury (e.g. MKK4, MLK, DLK, MKK4/7, JNK1/3 and SCG10) [69,81–84]. The MAPK cascade limits levels of Nmmat2 by selectively degrading membrane-associated, palmitoylated Nmnat2 [69,82].

Interestingly, Nmnat2 levels are differentially regulated. Both MAPK signalling and the atypical ubiquitin ligase complex are important for the fine-tuning of Nmnat2 levels. While MAPK signalling targets membrane-associated Nmnat2, the atypical ubiquitin ligase complex selectively degrades axoplasmic Nmnat2 [69,85]. The pharmacological inhibition of both mechanisms results in a strong attenuation of axon death signalling [69]. This suggests that distinct axonal pools of Nmnat2 are differentially regulated (figure 3a).

The above discoveries revealed a central and conserved function for Nmnat2/dNmnat. Levels of the labile axonal survival factor are dictated by three branches (figure 3a): (i) continuous supply of Nmnat proteins by axonal transport, (ii) constant degradation of vesicle-bound Nmnat proteins by the MAPK pathway, and (iii) constant degradation of soluble Nmnat proteins by the atypical ubiquitin ligase complex.

In healthy uninjured axons, Nmnat protein levels are above threshold, ensuring low NMN and high NAD$^+$ levels. Vice versa, in injured axons, the supply of Nmnat proteins by axonal transport is cut down, and, therefore, Nmnat protein degradation by the MAPK pathway and the atypical ubiquitin ligase complex takes over. This results in below-threshold levels of Nmnat proteins. Likewise, levels of NMN temporally rise, whereas NAD$^+$ levels drop.

So far, the execution of axon death signalling is solely initiated by below-threshold levels of Nmnat proteins. Any modification that sustains levels of Nmnat proteins—e.g. gain of Nmnat stability or loss of Nmnat protein degradation—ultimately attenuates axon death.

At this point, no mediator in axon death signalling had been identified that actively contributes to signalling, e.g. where loss-of-function mutations result in attenuated axon death signalling regardless of Nmnat protein levels. Below, we will discuss such mediators.

## 4.5. Sarm1/dSarm

The first discovery of a mutation which attenuates axon death irrespective of Nmnat levels was made in *Drosophila*: through an unbiased forward-genetic screen for axon death defective mutants, several loss-of-function alleles of the gene 'Drosophila sterile alpha and armadillo motif' (dsarm) were isolated (figure 3). Mutations in dsarm block axon death for the lifespan of the fly [86], dsarm is therefore essential for injury-induced axon degeneration. Similarly, mutants or downregulation of the mammalian homologue Sarm1 harbour a potent attenuation of axon death signalling in vitro and in vivo [76,86–88]. Sarm1/dSarm is a toll-like receptor adaptor family member and mainly expressed in the nervous system [86], yet it also functions in glial cells [89], and in the immune system [90].

Sarm1/dSarm contains three evolutionarily conserved protein domains: an Armadillo/HEAT (ARM) domain, a sterile alpha motif (SAM) (two in mammals and one in flies) and a Toll/interleukin-1 receptor homology (TIR) domain [86,87]. All three domains are essential for Sarm1 function in mice [76,82,87] and dSarm in *Drosophila* [65]. The ARM domain keeps dSarm/Sarm1 inactive, as previously reported in the *C. elegans* homologue TIR-1 [91], and the SAM domain is important for Sarm1 dimerization [76,81]. The TIR domain, rather than harbouring signalling activity, contains an enzymatic activity to consume NAD$^+$ (NADase activity) [92]. Sarm1 activation by dimerization is necessary and sufficient locally within axons to execute axon degeneration in the absence of injury. Its activation triggers rapid depletion of NAD$^+$ [73,76,92]. Importantly, the TIR domain NADase activity is evolutionarily conserved across flies, zebrafish and mice, where it cleaves NAD$^+$ into nicotinamide (Nam) and ADP-ribose (ADPR) or cyclic ADPR (cADPR) with species-specific differences [92]. These findings support a model where, upon injury, Sarm1/dSarm is activated and actively depletes NAD$^+$ levels in severed axons. Therefore, Sarm1/dSarm plays a central role in NAD$^+$ depletion in severed axons after injury.

## 4.6. Axundead

Recently, another essential mediator of axon death signalling has been identified in *Drosophila*. Several axundead (axed) mutants were isolated by another unbiased forward-genetic screen [65]. Similar to highwire and dsarm mutants, mutations in axed attenuate axon death signalling for the lifespan of the

fly (figure 3b) [20,65,86]. The *axed* gene consists of two isoforms (*axed^long* and *axed^short*), and both isoforms are capable of rescuing *axed* mutants. Axed proteins are predominantly found in axons and synapses, and its levels increase 4–6 h post axotomy and return to baseline 24 h after injury, suggesting either a transient change in localization and/or levels in response to axonal injury. Axed contains two evolutionary conserved domains: a BTB and a BACK domain, which suggests that Axed could dimerize and/or interact with cullin ubiquitin ligases [93]. There are four mammalian paralogues (BTBD1, BTBD2, BTBD3, BTBD6), and it remains to be determined which paralogue(s) could have an active role in axon death in mammals [65]. The precise mechanistic function of Axed remains currently unknown.

## 4.7. Calpains and Calpastatin

Calpains have recently also been shown to be involved in axon death signalling (figure 3a) [81]. Calpains are $Ca^{2+}$-dependent non-lysosomal proteases, and they are involved in neuronal degeneration in traumatic brain injury, cerebral ischaemia and AD [94]. Among the 15 mammalian isoforms, Calpain-1 and Calpain-2 are ubiquitously expressed, predominantly in the brain, and are present in both neurons and glia [95]. Within minutes after experimental traumatic axonal injury (TAI), axonal Calpain activity is elevated [96]. Vice versa, mice lacking Calpain-1/2 show a significant delay in axon death compared with wild-type mice [81].

Calpastatin is an endogenous *in vivo* inhibitor of Calpains. After axonal injury, levels of Calpastatin drop within 10 h, which correlates with the morphological degeneration of axons [34]. In contrast, transgenic mice expressing human Calpastatin harbour robust attenuation of axon death in transected optic nerves in the CNS and sciatic nerves in the PNS [97].

Calpastatin-mediated regulation of Calpains is likely the most downstream cascade of axon death signalling. However, the mechanistic link between the above mediators of axon death signalling and Calpains in mammals, and whether Calpains are also involved in other species, remains to be determined.

## 4.8. Other mediators

So far, axon death mediators were discussed whose modification results in a robust attenuation. Yet also other modifications have been reported that are capable of delaying axon death, such as the mitochondrial permeability transition [31], recycling endosomes [98], autophagy [99,100], sodium and potassium currents [50], the ubiquitin ligase ZNRF-1 [101], microtubule destabilization mediated by CRMP2 [102,103] and the transcription factor Pebbled/RREB1 [104]. The precise interaction of these mediators with the signalling cascade remains to be determined, because of either tissue-specific phenotypes or the involvement of whole organelles.

# 5. Interspecies commonalities and differences

Wld^S attenuates axon death in a variety of models across evolution [19,33,42–44,48]. Each animal system offers unique insights into axon death signalling, thus also distinct observations have been made between them. How is axon death studied in different animal models, and what is common, or different, among them? Below, we will briefly discuss each animal model where axon death is extensively studied, followed by highlighting commonalities and differences in axon death signalling across species.

## 5.1. Animal models

Mice are frequently used for both *in vitro* and *in vivo* axon death assays: *in vitro* cultured neurons from superior cell ganglia (SCGs) and dorsal root ganglia (DRGs) from the PNS, and retinal ganglion cells (RGCs) from the CNS, are subjected to axon death assays. Moreover, pharmacogenetic manipulation of metabolism can also be applied *in vitro*. Broadly used *in vivo* assays are optic nerve injuries (CNS) and sciatic nerve lesions (PNS).

Rats facilitate stereotaxic injections because of their larger brains, and lesions of optic or facial nerves and nerve roots can also be readily performed. Moreover, nerves are longer than those in mice, putting them closer to human axons, although human axons can still be 10-fold longer than their rat counterparts [24,44]. Rats also provide more abundant sources of tissue for biochemical and proteomic studies.

Zebrafish provides a major advantage with its powerful *in vivo* live imaging, as the complete time course of axon death, with the resolution of single axons, can readily be observed. It is important to mention that zebrafish offers the unique ability to visualize temporal $Ca^{2+}$ events *in vivo*.

Flies harbour the unique advantage of unbiased forward-genetic screens. Axon death is observed *in vivo* by antennal or maxillary palp ablation, wing injuries, and during larval development by nerve crush. These techniques were recently reviewed in [105]. Moreover, by the use of optogenetics, the functional preservation of axons and their synapses can readily be assessed by a simple grooming assay.

## 5.2. Commonalities and differences across species

Genetic analyses in the above animal systems provided important insights over recent years. These findings led to the definition of an axon death signalling pathway. It is tempting to combine all analyses to define a core signalling cascade across multiple species. Yet these analyses also revealed subtle differences that should not be neglected. Below, we will highlight common features as well as species-specific axon death signalling differences among mouse, fly and fish.

### 5.2.1. Conserved mediators

$Ca^{2+}$ plays a crucial role in all model systems tested so far. Both $Ca^{2+}$ entry into the axon after injury (first, rapid short-term $Ca^{2+}$ wave) and the $Ca^{2+}$ release from intracellular stores at the end of the lag phase (second, long-term $Ca^{2+}$ wave) have been observed across multiple species. It is therefore likely that the first conserved mediator of axon death signalling is external $Ca^{2+}$ influx into severed axons.

Nmnat2/dNmnat in mice, fish and flies harbour an evolutionary conserved feature too. As a labile axon survival factor, levels of Nmnat proteins matter: high or robust levels potently protect severed axons from undergoing axon death, and low levels induce axon degeneration in all species tested [33,45,60,64]. Alongside Nmnat, its biosynthetic $NAD^+$ activity is also conserved across species. It remains to be

determined whether levels of NMN and NAD$^+$, or ratios thereof, define when axon death is initiated, not only in mice but also in flies and fish.

Atypical ubiquitin ligase complex members, which regulate Nmnat protein levels, are conserved in mice and flies (Skip1a, Phr1, Fbxo45 in mouse, and Phr1 in fly). Interestingly, while the MAPK pathway plays an important role in mice, it seems to be negligible in flies [65]. Whether the fish homologues of the atypical ubiquitin ligase complex and the MAPK signalling cascade regulate levels of Nmnat2 in a similar way remains elusive.

The function of the dSarm/Sarm1 homologues in flies and mice, and Sarm1 in fish [106], is also evolutionarily conserved. Loss-of-function mutants block axon death signalling, and the TIR domain of these species harbours the ability to pathologically degrade NAD$^+$ to generate Nam and ADPR or cADPR [92].

It is equally important to note that species-specific pathway analyses revealed some remarkable differences. This indicates that we are far from fully understanding axon death signalling mechanisms. Below, we will briefly discuss these differences.

### 5.2.2. Differences across species

Mice lacking *Nmnat2* contain truncated axons during embryogenesis and die perinatally [63]. This lethality is partially rescued by over-expression of Wld$^S$ [107], and fully rescued by *Sarm1* mutants [88]. Therefore, Sarm1 executes axon death following below-threshold levels of Nmnat2. Sarm1 likely acts downstream of Nmnat2 (figure 3*a*). However, it could also act in parallel to it, but certainly not upstream of Nmnat2.

As soon as Nmnat2 levels drop below threshold, Sarm1 unleashes its TIR domain to consume NAD$^+$. Nmnat2, rather than maintaining NAD$^+$ levels through its NAD$^+$ biosynthetic activity, blocks NAD$^+$ consumption of Sarm1, which seems central to axon death signalling [73,76]. It remains completely unknown how Nmnat2 inhibits the NADase activity of Sarm1.

As mentioned above, *Nmnat2*$^{-/-}$ mice contain truncated axons during embryogenesis and die perinatally [63]. Wld$^S$ mice and *Sarm1* mutants are not the only two candidates able to rescue both: surprisingly, the expression of the bacterial NMN deamidase does the same in a dosage-dependent manner [75]. This finding suggests that, besides the pathological NAD$^+$ consumption of Sarm1, the temporal rise of NMN is also crucial for axon death signalling, after Nmnat2 has disappeared in severed axons (figure 3*a*).

In flies, sensory neurons mutant for the sole *dnmnat* gene undergo rapid neurodegeneration, which, unlike in mammals, is not blocked by *dsarm*, but by *axed* mutants [65]. These findings imply that neurodegeneration induced by below-threshold levels of dNmnat is not executed by dSarm, but by Axed. Similarly, axon and neurodegeneration induced by the expression of a constitutively active NADase version of dSarm lacking the inhibitory ARM domain (dSarm$^{\Delta ARM}$), which promotes rapid NAD$^+$ depletion, is blocked by *axed* mutants. This suggests that Axed is also downstream of dSarm. Finally, axon and neurodegeneration induced together by *dnmnat* mutants and by dSarm$^{\Delta ARM}$ expression is also blocked by *axed* mutants, suggesting that axon death signalling converges on Axed to execute the disassembly of the axon (figure 3*b*) [65].

In fish, axon death is attenuated by the over-expression of *nmnat2* and by Wld$^S$ [33,45]. Moreover, loss of *sarm1* also

attenuates axon death, suggesting that its function is conserved in fish [106]. However, it remains to be seen whether below-threshold levels of Nmnat2 are sufficient to trigger axon degeneration. Last but not least, the genetic interaction among Ca$^{2+}$, Nmnat2 and Sarm1 remains to be determined (figure 3*c*).

Above, we have summarized conserved axon death mediators, and we also highlighted differences observed across species. Despite these subtle differences, it is tempting to extract a core axon death mechanism which could be evolutionarily conserved. This will be discussed below.

### 5.3. The NMN/NAD$^+$ ratio

One speculative possibility is a crucial ratio between NMN and NAD$^+$. Under normal, healthy conditions, the ratio between NMN and NAD$^+$ is highly in favour of NAD$^+$ (NMN $\ll$ NAD$^+$), which is supported by the observation of lower axonal concentrations of NMN and higher concentrations of NAD$^+$ [73,74,88] (figure 3*a*). Injury leads to below-threshold levels of the labile axonal survival factor Nmnat2, which in turn leads to a temporal rise of NMN and lower levels of NAD$^+$, thus reducing the ratio (NMN < NAD$^+$). This ratio could already be sufficient to activate axon death. Importantly, below-threshold levels of Nmnat2 also discontinue Nmnat2-mediated inhibition of Sarm1, which discharges the NADase activity of Sarm1: the resulting Sarm1-mediated NAD$^+$ consumption tips the ratio towards NMN (NMN $\gg$ NAD$^+$). This ratio could therefore dictate whether axons should survive, or degenerate. Axon survival is favoured by NMN $\ll$ NAD$^+$: preventing the rise of NMN, or the consumption of NAD$^+$, both potently attenuate axon death. Vice versa, conditions altering this ratio will induce axon death: the forced depletion of NAD$^+$ by a constitutively active NADase activity of Sarm1/dSarm is sufficient to trigger axon (and cell body) death. If this were true, high NMN levels—significantly higher than NAD$^+$—should also be capable of triggering axon death *in vivo*. This remains to be tested, but also harbours technical difficulties: Nmnat proteins efficiently synthesize NAD$^+$ from NMN, thus a rise in NMN automatically results in a rise of NAD$^+$ [58,73]. Interestingly, high levels of a cell-permeable analogue of NMN (which cannot be used as substrate for Nmnat-mediated NAD$^+$ synthesis) are sufficient to activate Sarm1 in cultured neurons, thereby depleting NAD$^+$, which results in non-apoptotic cell death. Whether this is also true in severed axons remains to be determined [108].

We are still far from understanding why and how axons execute their own disassembly. Our knowledge of axon death as an emerging signalling cascade is still in its infancy. Each axon death mediator revealed crucial mechanistic insights over recent years, and helped to define Sarm1 as a first target that can be translated to the clinic. In the section below, we will briefly discuss where targeting of Sarm1 is beneficial in mouse models of neurological conditions.

## 6. Axon death signalling in disease

Mutations in human Nmnat2 have been linked to fetal akinesia deformation sequence (FADS), and to childhood-onset polyneuropathy and erythromelalgia [109,110]. These discoveries provide the first direct molecular evidence that axon death in WD is involved in a human axonal disorder. They support

royalsocietypublishing.org/journal/rsob    Open Biol. 9: 190118

royalsocietypublishing.org/journal/rsob Open Biol. 9: 190118

the idea that axonopathies are a major contributor to certain human neurodegenerative disorders [12–15].

Wld$^S$ and its capacity to slow down injury-induced axon degeneration offered the first opportunity to attenuate axonopathies which occur in the absence of injury. Over recent years, distinct outcomes were observed in a broad range of disease models, including extended lifespan, age-dependent effects, improved performance, or a lack of amelioration [23]. The promising neuroprotective function of Wld$^S$ also led to efforts developing drugs to stabilize Nmnat proteins [111]. However, while the efficacy of Wld$^s$ or any other form of engineered Nmnat protein seems promising in certain models of neurological conditions, therapeutic potential may be limited due to a gain-of-function protein: its efficacy relies on dosage, which could be challenging during long-term therapeutic treatments [43]. Moreover, Wld$^S$ harbours age-dependent effects [112], and while it is capable of slowing down axon death, it is not able to fully block it [107]. Therefore, alternative or complementary approaches would increase the chance of ameliorating axonopathies.

The manipulation of the NMN/NAD$^+$ ratio could offer an attractive alternative. It could be achieved by pharmacological targeting of specific metabolic pathways, or by simple oral supplement of NAD$^+$ or its precursors. Preventing either NAD$^+$ consumption or a temporal rise of NMN could serve as attractive strategies [113,114]. However, without any specificity towards neuronal tissue, the manipulation of NAD$^+$ metabolism also bears great risks [115].

The discovery of Sarm1/dSarm has offered a novel and unique therapeutic opportunity. In order to block axon death signalling, Sarm1 protein or its activity has to be decreased, rather than increased such as with Nmnat2. Like Wld$^S$, Sarm1 mutant mice offer a range of outcomes in mouse models of neurological conditions:

When axons are challenged by mechanical forces, Sarm1$^{-/-}$ mice harbour reduced neurological deficits and a better functional outcome in a model of TBI [116]. Sarm1$^{-/-}$ mice also contain significantly reduced axonal lesions in a model of TAI with impact acceleration of heads [117]. In addition, an *in vivo* gene therapy approach using adeno-associated virus to deliver a dominant-negative version of Sarm1 had similar effects in an injury-induced axon degeneration model to that in Sarm1$^{-/-}$ mice [118]. These findings strongly suggest that Sarm1 serves as a promising therapeutic target to ameliorate force-induced axonopathies.

In two models of CIPN (e.g. vincristine or paclitaxel), and in a model of metabolic-induced peripheral neuropathy, mice lacking Sarm1 prevent the distal degeneration of myelinated axons and electrophysiological abnormalities [119,120]. These findings fuel hope that axonopathies caused by chemotherapy or by diabetes—the most common causes of peripheral neuropathies [121,122]—could also be therapeutically targeted in patients.

However, while Sarm1 harbours the potential to ameliorate certain axonopathies, it is important to note that the lack of Sarm1 does not suppress motor neuron degeneration in a mouse model of ALS [123]. It remains to be determined whether Sarm1 is dispensable in other ALS mouse models.

So far, Sarm1$^{-/-}$ mutants have beneficial effects in certain models of neurological conditions. This raises hope that Sarm1 and other axon death mediators could serve as druggable targets to halt axon loss. Axon death signalling serves therefore as an attractive pathway to develop therapies against, with the ultimate goal to prevent and treat axon loss in a broad range of neurological diseases.

# 7. Conclusion

The discovery and characterization of axon death signalling, which is activated by injury, not only provided exciting insights into the underlying mechanism mediating axonal self-destruction after injury. It also revealed that the axon death pathway is highjacked in other challenging conditions for the nervous system where axonopathies occur in the absence of injury. Therefore, it is crucial to fill missing mechanistic gaps among already identified axon death mediators, and to identify other key mediators required for axon death signalling. This will help to expand our understanding of a signalling pathway that ultimately leads to the death of the axon. Overall, there is certainly a lot left to do, and axon death research will therefore continue to be a lively field in the future.

Data accessibility. This article has no additional data.

Competing interests. The authors declare that they have no competing interests.

Funding. This work is supported by a Swiss National Science Foundation (SNSF) Assistant Professor Grant (176855), the État de Vaud (University of Lausanne) and the International Foundation for Research in Paraplegia (IRP) (P180) to L.J.N.

Acknowledgements. The authors thank Dr Tilmann Achsel and Maria Paglione for helpful comments.

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
