## [Reviewer comments · Open Biology]

Review History

RSOB-19-0118.R0 (Original submission)

Review form: Reviewer 1

Recommendation

Major revision is needed (please make suggestions in comments)

Scientific importance: Is the manuscript an original and important contribution to its field?

Acceptable

General interest: Is the paper of sufficient general interest?

Acceptable

Quality of the paper: Is the overall quality of the paper suitable?

Marginal

It is a condition of publication that authors make their supporting data, code and materials available - either as supplementary material or hosted in an external repository. Please rate, if applicable, the supporting data on the following criteria.

Is it accessible?

N/A

Is it clear?

N/A

Is it adequate?

N/A

Do you have any ethical concerns with this paper?

No

Comments to the Author

The title of this review suggests a broad overview of the molecular mechanisms of the major forms of axonal degeneration with a special focus on different species and diseases. However, in the present manuscript only Wallerian degeneration (WD) is discussed in detail while other forms of axonal degeneration are not introduced. WD is not put in a broader context. Moreover, the role of axonal degeneration (or WD in particular) in disease is described very incompletely on only 1 page.

Therefore either the title should be adjusted to e.g. "Axon death signalling in Wallerian degeneration among different species" or the manuscript should be significantly expanded to include other forms of axonal degeneration with the respective molecular mechanisms and a much broader and more thorough discussion about the role of axonal degeneration in disease.

WD is only one form of axonal degeneration (others including acute axonal degeneration, dying back axonal degeneration, focal axonal degeneration, retraction, pruning etc.). This should be made very clear and either the other forms of axonal degeneration should be discussed in detail or the article should be merely focussed on WD.

WD affects only the distal part of the axon after a lesion but not the remaining proximal part that is still connected to the soma. In many cases, especially after neurotrauma, it is a necessary prerequisite for later axonal regeneration that the distal part of the degenerating axon is cleared. It is therefore rather the remaining proximal axon stump that should be stabilised as it is still connected to the soma and might be the origin of axonal regeneration. WD in this case is a necessary physiological process that should not be inhibited.

In line with this view, several studies have shown that Wld-S counteracts and significantly delays axonal regeneration. These studies should also be discussed here (e.g. overview/perspective: Tang, Cells 2019 "Why is NMNAT Protective against Neuronal Cell Death and Axon Degeneration, but Inhibitory of Axon Regeneration?" and many others).

Autophagy has been described as an important pathophysiological mediator of WD and should thus be discussed in the review (e.g. Wang et al., Sci Adv 2019 "Rapid depletion of ESCRT protein Vps4 underlies injury-induced autophagic impediment and Wallerian"; Wakatsuki et al., JCB 2017 "GSK3B-mediated phosphorylation of MCL1 regulates axonal autophagy to promote Wallerian degeneration").

For the discussion of calcium as a "death mediator", the following studies should be cited as they have nicely proven the mechanism in mammals in vivo: Kerschensteiner et al., Nat Med 2005 "In

vivo imaging of axonal degeneration and regeneration in the injured spinal cord." and Knöferle et al., PNAS 2010 "Mechanisms of acute axonal degeneration in the optic nerve in vivo".

The involvement of calpain in axonal degeneration has also been demonstrated in mammals in vivo and here CRMP2 could be identified as a major downstream target: Zhang et al., Sci Rep 2016 "Calpain-mediated cleavage of collapsin response mediator protein-2 drives acute axonal degeneration". This should be included.

If the authors really wish to discuss the role of WD in neurological diseases, this part needs to be substantially expanded. Also negative results of therapeutic interventions in WD should be discussed, e.g. the negative effects on axonal regeneration and negative results in MS models (e.g. Singh et al., 2017 "Relationship of acute axonal damage, Wallerian degeneration, and clinical disability in multiple sclerosis"). Also differential effects on axon and soma should be taken into account (e.g. Beirowski Eur J Nsc 2008 "The WldS gene delays axonal but not somatic degeneration in a rat glaucoma model").

Figure 1 does not include the events of acute axonal degeneration and of possible dying back axonal degeneration of the proximal axon. This should at least be mentioned in the legend. Also autophagy as a major player in axonal degeneration and clearing of axons should be included in the figure. It would be desirable to include NMNAT as the major molecular player in WD in the figure.

Decision letter (RSOB-19-0118.R0)

15-Jul-2019

Dear Professor Neukomm

We are pleased to inform you that your manuscript RSOB-19-0118 entitled "Axon death signaling in species and disease" has been accepted by the Editor for publication in Open Biology. The reviewer has recommended publication, but also suggest some minor revisions to your manuscript. Therefore, we invite you to respond to the comments and revise your manuscript.

Please submit the revised version of your manuscript within 7 days. If you do not think you will be able to meet this date please let us know immediately and we can extend this deadline for you.

When submitting your revised manuscript, you will be able to respond to the comments made by the referee and upload a file "Response to Referees" in "Section 6 - File Upload". You can use this to document any changes you make to the original manuscript. In order to expedite the processing of the revised manuscript, please be as specific as possible in your response to the referee.

- 1) A text file of the manuscript (doc, txt, rtf or tex), including the references, tables (including captions) and figure captions. Please remove any tracked changes from the text before submission. PDF files are not an accepted format for the "Main Document".
- 2) A separate electronic file of each figure (tiff, EPS or print-quality PDF preferred). The format should be produced directly from original creation package, or original software format. Please note that PowerPoint files are not accepted.
- 3) Electronic supplementary material: this should be contained in a separate file from the main text and meet our ESM criteria (see <http://royalsocietypublishing.org/instructions-authors#question5>). All supplementary materials accompanying an accepted article will be treated as in their final form. They will be published alongside the paper on the journal website and posted on the online figshare repository. Files on figshare will be made available approximately one week before the accompanying article so that the supplementary material can be attributed a unique DOI.

Online supplementary material will also carry the title and description provided during submission, so please ensure these are accurate and informative. Note that the Royal Society will not edit or typeset supplementary material and it will be hosted as provided. Please ensure that the supplementary material includes the paper details (authors, title, journal name, article DOI). Your article DOI will be 10.1098/rsob.2016[*last 4 digits of e.g. 10.1098/rsob.20160049*].

- 4) A media summary: a short non-technical summary (up to 100 words) of the key findings/importance of your manuscript. Please try to write in simple English, avoid jargon, explain the importance of the topic, outline the main implications and describe why this topic is newsworthy.

Images

Data-Sharing

It is a condition of publication that data supporting your paper are made available. Data should be made available either in the electronic supplementary material or through an appropriate repository. Details of how to access data should be included in your paper. Please see <http://royalsocietypublishing.org/site/authors/policy.xhtml#question6> for more details.

Data accessibility section

Sincerely,

The Open Biology Team
 mailto:openbiology@royalsociety.org

Editor's comment: Many of the reviewer's comments can be addressed by changing the title of the article to a more specific one. I suggest doing this. Please then address remaining comments and indicate in a cover letter how you have done so. Thanks

Reviewer's Comments to Author:

Referee:

Comments to the Author(s)

The title of this review suggests a broad overview of the molecular mechanisms of the major forms of axonal degeneration with a special focus on different species and diseases. However, in the present manuscript only Wallerian degeneration (WD) is discussed in detail while other forms of axonal degeneration are not introduced. WD is not put in a broader context. Moreover, the role of axonal degeneration (or WD in particular) in disease is described very incompletely on only 1 page.

Therefore either the title should be adjusted to e.g. "Axon death signalling in Wallerian degeneration among different species" or the manuscript should be significantly expanded to include other forms of axonal degeneration with the respective molecular mechanisms and a much broader and more thorough discussion about the role of axonal degeneration in disease.

WD is only one form of axonal degeneration (others including acute axonal degeneration, dying back axonal degeneration, focal axonal degeneration, retraction, pruning etc.). This should be made very clear and either the other forms of axonal degeneration should be discussed in detail or the article should be merely focussed on WD.

WD affects only the distal part of the axon after a lesion but not the remaining proximal part that is still connected to the soma. In many cases, especially after neurotrauma, it is a necessary prerequisite for later axonal regeneration that the distal part of the degenerating axon is cleared. It is therefore rather the remaining proximal axon stump that should be stabilised as it is still connected to the soma and might be the origin of axonal regeneration. WD in this case is a necessary physiological process that should not be inhibited.

In line with this view, several studies have shown that Wld-S counteracts and significantly delays axonal regeneration. These studies should also be discussed here (e.g. overview/perspective: Tang, Cells 2019 "Why is NMNAT Protective against Neuronal Cell Death and Axon Degeneration, but Inhibitory of Axon Regeneration?" and many others).

Autophagy has been described as an important pathophysiological mediator of WD and should thus be discussed in the review (e.g. Wang et al., Sci Adv 2019 "Rapid depletion of ESCRT protein Vps4 underlies injury-induced autophagic impediment and Wallerian"; Wakatsuki et al., JCB 2017 "GSK3B-mediated phosphorylation of MCL1 regulates axonal autophagy to promote Wallerian degeneration").

For the discussion of calcium as a "death mediator", the following studies should be cited as they have nicely proven the mechanism in mammals in vivo: Kerschensteiner et al., Nat Med 2005 "In

vivo imaging of axonal degeneration and regeneration in the injured spinal cord.” and Knöferle et al., PNAS 2010 “Mechanisms of acute axonal degeneration in the optic nerve in vivo”.

The involvement of calpain in axonal degeneration has also been demonstrated in mammals in vivo and here CRMP2 could be identified as a major downstream target: Zhang et al., Sci Rep 2016 “Calpain-mediated cleavage of collapsin response mediator protein-2 drives acute axonal degeneration”. This should be included.

If the authors really wish to discuss the role of WD in neurological diseases, this part needs to be substantially expanded. Also negative results of therapeutic interventions in WD should be discussed, e.g. the negative effects on axonal regeneration and negative results in MS models (e.g. Singh et al., 2017 “Relationship of acute axonal damage, Wallerian degeneration, and clinical disability in multiple sclerosis”). Also differential effects on axon and soma should be taken into account (e.g. Beirowski Eur J Nsc 2008 “The WldS gene delays axonal but not somatic degeneration in a rat glaucoma model”).

Figure 1 does not include the events of acute axonal degeneration and of possible dying back axonal degeneration of the proximal axon. This should at least be mentioned in the legend. Also autophagy as a major player in axonal degeneration and clearing of axons should be included in the figure. It would be desirable to include NMNAT as the major molecular player in WD in the figure.

Author's Response to Decision Letter for (RSOB-19-0118.R0)

See Appendix A.

Decision letter (RSOB-19-0118.R1)

02-Aug-2019

Dear Professor Neukomm

We are pleased to inform you that your manuscript entitled "Axon death signaling in Wallerian degeneration among species and in disease" has been accepted by the Editor for publication in Open Biology.

Sincerely,

The Open Biology Team
mailto: openbiology@royalsociety.org

Appendix A

Point-by-point response to the reviewer's comments (manuscript RSOB-19-0118)

We would like to thank the reviewer for comments, suggestions and concerns. Below, we will respond point-by-point to each comment raised by the reviewer. In the manuscript as well as in this document, changes are highlighted in red. In this document, the grey text serves as help for orientation.

Comments to the Author(s)

The title of this review suggests a broad overview of the molecular mechanisms of the major forms of axonal degeneration with a special focus on different species and diseases. However, in the present manuscript only Wallerian degeneration (WD) is discussed in detail while other forms of axonal degeneration are not introduced. WD is not put in a broader context. Moreover, the role of axonal degeneration (or WD in particular) in disease is described very incompletely on only 1 page. Therefore either the title should be adjusted to e.g. "Axon death signalling in Wallerian degeneration among different species" or the manuscript should be significantly expanded to include other forms of axonal degeneration with the respective molecular mechanisms and a much broader and more thorough discussion about the role of axonal degeneration in disease.

The focus of this review lies in axon death signaling in Wallerian degeneration (WD). In the first part of the manuscript, we defined WD as a two-step process consisting of axon death and glial clearance. As the reviewer correctly indicated, we specifically discuss axon death in WD, while other forms of axon degeneration weren't in the main focus of this review. We therefore adjusted the title to:

Axon death signaling in Wallerian degeneration among species and in disease

We would like to keep "disease" in the title, for the following two reasons:

1) Wld^S serves as a golden standard in the field of WD. Since its discovery, Wld^S has been thoroughly used in a broad range of models with neurological conditions (*C. elegans*, *Drosophila* and mouse). If axon degeneration occurs in the absence of injury, and can be blocked by over-expression of Wld^S, they undergo Wallerian-like degeneration. A beneficial outcome mediated by over-expression of Wld^S led to the conclusion that the signaling pathway of WD (axon death) might be engaged in those axons. An important aspect of Wld^S is dosage. The stronger it is over-expressed, the better it protects, which suggests that levels of Wld^S dictate whether there might be a beneficial outcome. To date, multiple reviews have already discussed how attenuated axon death signaling in WD (that is, over-expression of Wld^S) might be beneficial in axons in a broad range of diseased and injured nervous systems, and we included (Conforti *et al.* 2014, Nature Neuroscience) in the chapter "6. Axon death signaling in disease". This was the main reason for us to emphasize in the manuscript that Wld^S will not be covered as representative of the WD signaling pathway in disease.

2) Since the discovery of *dSarm/Sarm1* in 2012, several labs started to investigate the role of *Sarm1* in the diseased and injured nervous system. Contrary to the gain-of-function situation in Wld^S, loss of *dSarm/Sarm1* has a much more potent attenuation phenotype than Wld^S (Gilley *et al.*, 2017, Cell Reports). This is why we focused on *Sarm1*, which, as indicated above, offers 1 page of insights into the diseased and injured nervous system in our review. Importantly, we also emphasize that *Sarm1*^{-/-} knockout mice are not protected in a model of ALS (SOD1 G93A), while Wld^S offers modest extension of life span. Given that *Nmnat/Nmnat2* (Wld^S) controls *dSarm/Sarm1* activation, this is unexpected. And it shows that over-expression of Wld^S might do something to axons which is unrelated to axon death signaling in WD.

We therefore believe that Wld^S should not be the sole reference for axon death signaling in WD. For this reason, we mainly focused on Sarm1 in mouse models of neurological conditions.

WD is only one form of axonal degeneration (others including acute axonal degeneration, dying back axonal degeneration, focal axonal degeneration, retraction, pruning etc.). This should be made very clear and either the other forms of axonal degeneration should be discussed in detail or the article should be merely focused on WD.

We added a new section which clarifies that among the different forms of axon degeneration, we will focus on axon death signaling in WD:

To date, distinct morphological modes of axon degeneration have been observed and underlying molecular mechanisms described [10]. Among them are dying back axon degeneration, retraction, axosome shedding, focal axonal degeneration induced by growth factor withdrawal and pruning, to name a few. Axon degeneration can also be triggered through axonal injury (axotomy), which is probably one of the simplest models to study how axons execute their own destruction. Identified by and named after Augustus Waller, Wallerian degeneration (WD) is...

...

Over recent years, significant advancements broadened our understanding of axon death signaling in WD.

WD affects only the distal part of the axon after a lesion but not the remaining proximal part that is still connected to the soma. In many cases, especially after neurotrauma, it is a necessary prerequisite for later axonal regeneration that the distal part of the degenerating axon is cleared. It is therefore rather the remaining proximal axon stump that should be stabilised as it is still connected to the soma and might be the origin of axonal regeneration. WD in this case is a necessary physiological process that should not be inhibited.

In line with this view, several studies have shown that Wld-S counteracts and significantly delays axonal regeneration. These studies should also be discussed here (e.g. overview/perspective: Tang, Cells 2019 "Why is NMNAT Protective against Neuronal Cell Death and Axon Degeneration, but Inhibitory of Axon Regeneration?" and many others).

We are aware of the overview/perspective mentioned by the reviewer (Tang, Cells 2019) and other related reviews. In this particular review/perspective, there are multiple studies covered that show how the proximal axon (connected to the soma) induces stress-responses, and ultimately engages axon regeneration. We are also aware that attenuated axon death signaling harbors inhibitory features required for axons to regenerate. However, we believe that axon regeneration would be a topic for a separate review. We therefore avoided discussing the proximal axon (e.g. axon regeneration), and focused specifically on the distal from the soma separated axon (e.g. axon degeneration).

Moreover, the message of this review is not to suggest that WD, as a necessary physiological process, should be inhibited after injury in the distal, from the soma separated axon. Rather, it provides an overview of essential genes required for axon death signaling in WD.

Autophagy has been described as an important pathophysiological mediator of WD and should thus be discussed in the review (e.g. Wang et al., Sci Adv 2019 "Rapid depletion of ESCRT protein Vps4

underlies injury-induced autophagic impediment and Wallerian"; Wakatsuki et al., JCB 2017 "GSK3B-mediated phosphorylation of MCL1 regulates axonal autophagy to promote Wallerian degeneration").

We agree with the reviewer's comment. In fact, some work was already mentioned and cited under chapter "4.8. Other mediators". We significantly expanded this paragraph which now reads as follows:

So far, axon death mediators were discussed whose modification results in a robust attenuation. Yet also other modifications have been reported that are capable to delay axon death, **such as the mitochondrial permeability transition [31], recycling endosomes [98], autophagy [99,100], sodium and potassium currents [50], the ubiquitin ligase ZNRF-1 [101], the microtubule destabilization mediated by CRMP2 [102,103] and the transcription factor Pebbled/RREB1 [104]**. The precise interaction of these mediators with the signaling cascade remains to be determined, either due to tissue specific phenotypes or the involvement of whole organelles.

For the discussion of calcium as a "death mediator", the following studies should be cited as they have nicely proven the mechanism in mammals *in vivo*: Kerschensteiner et al., Nat Med 2005 "In vivo imaging of axonal degeneration and regeneration in the injured spinal cord." and Knöferle et al., PNAS 2010 "Mechanisms of acute axonal degeneration in the optic nerve *in vivo*".

We added both references under chapter "4.1. Ca²⁺":

As mentioned above, a local initial influx of Ca²⁺ right after axotomy from the site of injury is observed both in invertebrate and vertebrate models *in vivo* and *in vitro* [24,27,50,51]. The first, rapid short-term Ca²⁺ wave precedes the lag phase (Figure 2). The disassembly of severed axons is initiated by high extra-axonal Ca²⁺ concentrations: if the extracellular environment is devoid of Ca²⁺ (by adding EGTA, or in a medium lacking Ca²⁺), or when voltage-gated Ca²⁺ channels are inhibited, the fragmentation of severed axons is significantly delayed (Figure 3A) [26,50,52–54].

The involvement of calpain in axonal degeneration has also been demonstrated in mammals *in vivo* and here CRMP2 could be identified as a major downstream target: Zhang et al., Sci Rep 2016 "Calpain-mediated cleavage of collapsin response mediator protein-2 drives acute axonal degeneration". This should be included.

As mentioned above, we significantly expanded the chapter "4.8. Other mediators". One CRMP2 reference was already included in our manuscript (Kinoshita *et al.*, 2019), we now added a second reference (Zhang *et al.*, 2016), both of which are highlighting CRMP2:

So far, axon death mediators were discussed whose modification results in a robust attenuation. Yet also other modifications have been reported that are capable to delay axon death, **such as the mitochondrial permeability transition [31], recycling endosomes [98], autophagy [99,100], sodium and potassium currents [50], the ubiquitin ligase ZNRF-1 [101], microtubule destabilization mediated by CRMP2 [102,103] and the transcription factor Pebbled/RREB1 [104]**. The precise interaction of these mediators with the signaling cascade remains to be determined, either due to tissue specific phenotypes or the involvement of whole organelles.

If the authors really wish to discuss the role of WD in neurological diseases, this part needs to be substantially expanded. Also negative results of therapeutic interventions in WD should be discussed, e.g. the negative effects on axonal regeneration and negative results in MS models (e.g. Singh et al., 2017 “Relationship of acute axonal damage, Wallerian degeneration, and clinical disability in multiple sclerosis”). Also differential effects on axon and soma should be taken into account (e.g. Beirowski Eur J Nsc 2008 “The WldS gene delays axonal but not somatic degeneration in a rat glaucoma model”).

We appreciate the above comments raised by the reviewer. One concerns WD in neurological diseases, the other one the effects in axons vs. soma.

Regarding WD in disease: as mentioned above, Wld^S has been thoroughly tested in a broad range of models with neurological conditions with different outcomes, which is covered in several reviews. Wld^S (or Nmnat2) regulates the activation of Sarm1 as an executioner of axon death signaling in WD. Axon death can therefore be attenuated by over-expression of Wld^S (which depends on dosage), or lack of Sarm1. *Sarm1*^{-/-} knockout mice circumvent dosage issues. Therefore, we wanted to summarize the role of Sarm1 as a pro-axon death gene in disease models. Sarm1 has only been recently investigated in disease models, and therefore only a few studies are available. Most, if not all, are covered in one page in the chapter “6. Axon death signaling in disease”. Importantly, we also included negative results (e.g. Sarm1 in an ALS mouse model).

Regarding axon vs. soma: JNK signaling, Sarm1 and other axon death genes have been implicated in stress response in the soma of the neuron after injury (Wang *et al.*, 2018, Cell Reports). However, as indicated above, the goal of this review is to focus on the distal axon, and not on the proximal axon or the soma itself, which could be covered in a separate review.

Figure 1 does not include the events of acute axonal degeneration and of possible dying back axonal degeneration of the proximal axon. This should at least be mentioned in the legend. Also autophagy as a major player in axonal degeneration and clearing of axons should be included in the figure. It would be desirable to include NMNAT as the major molecular player in WD in the figure.

As mentioned above, in this review, we would like to focus on the distal, from the soma separated axon. Therefore, we avoided the proximal axon, including the discussion of a possible dying back mechanism after injury.

Moreover, rather than focusing on specific molecules or programs in Figure 1 (e.g. Nmnat/Nmnat2 or autophagy, respectively), we wanted to provide a broad overview about WD, that is the axonal intrinsic axon death signaling cascade, and the glial extrinsic clearance machinery. Furthermore, Nmnat/Nmant2 is discussed in details later in the manuscript, and indicated in Figure 3.

Additional changes:

1. We would like to brief both the editor and the reviewer that some minor changes were added to the manuscript (highlighted in red), e.g. educt is changed to **substrate**, among is changed to **between**, etc.

2. We also changed two citations from bioRxiv to pubmed references:

Mutations in human Nmnat2 have been linked to Fetal Akinesia Deformation Sequence (FADS), and to childhood onset polyneuropathy and erythromelalgia [110,111].

3. In addition, we now included two minor updates.

- The first update includes an active role of glia in axonal fragmentation:

During the execution phase, surrounding glial cells and specialized phagocytes **not only** clear the resulting axonal debris **by activating multiple signaling pathways** [19,36–39], **but also actively enhance axonal fragmentation** [40]. To date, several **key** questions remain unresolved, such as what kind of axonal “eat-me” signals are presented to surrounding glia [41], and where within the execution phase **glial cells** engage to clear axonal debris.

Once the execution phase is over, e.g. when the **injured** axon is disassembled and the resulting debris cleared by surrounding glia, WD is also over. **While WD occurs both in the central and peripheral nervous system (CNS and PNS, respectively), it is of particular importance in the PNS** to pave the way for the proximal **axon still attached to the neuronal soma**, which **ultimately** activates its regenerative program, to regrow and thereby to re-establish circuit function [33].

- The second update includes the role of NMN in Sarm1 activation:

Nmnat proteins efficiently synthesize NAD⁺ from NMN, thus a raise in NMN automatically results in a raise of NAD⁺ [58,73]. **Interestingly, high levels of a cell-permeable analogue of NMN (which cannot be used as substrate for Nmnat-mediated NAD⁺ synthesis) are sufficient to activate Sarm1 in cultured neurons, thereby depleting NAD⁺, which results in non-apoptotic cell death. Whether this is also true in severed axons remains to be determined [109].**

4. Following the author guidelines available on the royal society webpage, we changed the format of the bibliography from Vancouver to Open Biology style. This format change did not affect the content from the original manuscript.